# CMOS Image Sensors and Plasma Processes: How PMD Nitride Charging Acts on the Dark Current

**DOI:** 10.3390/s19245534

**Published:** 2019-12-14

**Authors:** Yolène Sacchettini, Jean-Pierre Carrère, Romain Duru, Jean-Pierre Oddou, Vincent Goiffon, Pierre Magnan

**Affiliations:** 1STMicroelectronics, 850 rue Jean Monnet, 38920 Crolles, France; yolene.sacchettini@st.com (Y.S.);; 2ISAE-SUPAERO, Université de Toulouse, 10 av Edouard Belin, 31055 Toulouse, France; vincent.goiffon@isae-supaero.fr (V.G.); pierre.magnan@isae-supaero.fr (P.M.)

**Keywords:** plasma induced damage, CMOS image sensor, nitride charging, PMD stack, dark current

## Abstract

Plasma processes are known to be prone to inducing damage by charging effects. For CMOS image sensors, this can lead to dark current degradation both in value and uniformity. An in-depth analysis, motivated by the different degrading behavior of two different plasma processes, has been performed in order to determine the degradation mechanisms associated with one plasma process. It is based on in situ plasma-induced charge characterization techniques for various dielectric stack structures (dielectric nature and stack configuration). A degradation mechanism is proposed, highlighting the role of ultraviolet (UV) light from the plasma in creating an electron hole which induces positive charges in the nitride layer at the wafer center, and negative ones at the edge. The trapped charges de-passivate the SiO_2_/Si interface by inducing a depleted interface above the photodiode, thus emphasizing the generation of dark current. A good correlation between the spatial distribution of the total charges and the value of dark current has been observed.

## 1. Introduction

The dark noise of CMOS image sensors can be affected by plasma process steps, either by purely electrical stress [1] or by the combination of ultraviolet (UV) photon interaction and electrical stress [2]. This dark noise degradation is caused by an increase of the dark current non-uniformity into the pixel matrix, or a temporal noise degradation of the pixel MOS. This plasma-induced damage occurring during the sensor process can cause severe yield loss, or even pixel reliability issues. We have isolated here an oxygen plasma used as a dry strip step as the origin of dark current degradation. We have studied how the dark current on a front side illumination (FSI) sensor is linked to the interface damages and the nature of the charges appearing in the pre-metal dielectrics (PMD) stack during the plasma exposition. To characterize the impact of such plasma processes on these dielectrics, the potentials, charge and interface states are characterized using a capacitive Kelvin probe associated with the corona oxide characterization of semiconductor (COCOS) technique [3]. Then, TCAD simulation will illustrate the link between dark current and the results on the dielectrics characterization.

## 2. Dark Current Degradation by Plasma Strip Process

### 2.1. Experimental

The device studied is a FSI CMOS image sensor with 4.1 µm pitch, 4T pinned N-photodiode pixels. The pixels are physically isolated by a deep trench isolation (DTI). To improve the optical path and the quantum efficiency of the pixel, a cavity is etched in the dielectrics of the back end of line (BEOL), before the color filter and the microlens patterning. The pixel configuration is given in Figure 1a, where cross section of both the active pixels and the dark reference pixels, shielded by a metal plate, are shown. The Silicon-Carbon-Nitride (SiCN) layers between the photodiode and the cavity are also etched in our standard process, to prevent optical interferences. In the following experiences, these SiCN layers have been varied from three to none, as illustrated on Figure 3. Then, two processes to strip the resist after the cavity etch are compared. The first one is a high density oxygen plasma made by a magnetic enhanced etch reactor and referred to as “strip A”. The second is done in a dedicated stripping tool with a remote plasma, referred to as “strip B”. These two processes are illustrated in Figure 2b.

### 2.2. Pixel Dark Current Results

Figure 1b shows the mean dark current distribution on a wafer, for both the active and the dark reference pixel populations, and processed with strip “A”. We observe that the dark reference pixels have a low dark current, and are very uniform across the wafer, whereas the active pixels show degraded dark current, and are very dispersed on the wafer. The metal shield of the dark pixel seems to have prevented damage.

Figure 2 show a significant increase of the active pixel dark current, while “strip B” process prevents any damage, matching the dark current level of the reference dark pixels. This indicates that the “strip A” process is the cause of the degradation. It will be used as a study case thereafter.

The dark current degradation of the active pixels decreases with the presence of SiCN layers, as indicated in Figure 3e. Two SiCN layers above the photodiode allow the pixel to be protected against the plasma induced damage. Moreover, the wafer mapping of the dark current shown in Figure 3f indicates a strong center to edge of the wafer non uniformity, with the highest dark current values observed on the center.

Furthermore, there is no influence on damage of the pixel design, like the deep or shallow trench perimeter, or the poly transfer gate length, as shown in Figure 4. As a consequence, the high dark current source on degraded pixel seems not to be located on the pixel periphery. Therefore, we can assume that the dark current shows a pure area-dependent behavior, and so that the degradation origin mainly comes from the interface above the photodiode, under the PMD stack.

### 2.3. Degradation Mechanism Hypothesis

The interactions between a plasma process and some device under the wafer surface are numerous, and come from the plasma properties, as illustrated in Figure 5. First, a plasma imposes an electric field in the sheath between its periphery and all the floating surfaces. This is due to the velocity differences between the positive ions and the faster electrons. This electric field into the sheath and the plasma potential can be non-uniform across the wafer, and this causes some voltage differences between the conductive wafer substrate and the floating wafer surface. Then, plasma processes also generate some deep ultraviolet (UV) radiation due to the energetic collisions between electrons and neutrons.

Each of these two elements may cause some device damage. First, the voltage at the wafer surface is well known [4] to generate gate oxide damage by the antenna charging effect. We have demonstrated elsewhere [5] that such oxygen plasma can degrade a 5 nm MOS gate oxide by antenna voltage reaching about 9–10 V. Here, such an antenna effect is not probable, because the damage occurs during a dielectric etch and strip process: there is no conductive area exposed to the plasma. Moreover, the dark current damage does not depend on the transfer gate oxide area.

Secondly, the UV radiation from the plasma can generate charge in the nitride [6] or in the silicon oxide [7] and can increase conductivity in the dielectrics [8].

In our case, due to the fact that the presence of SiCN layers modulate the degradation and that shielded pixels are unaffected, it is unlikely that a pure electrical stress is causing the charge trapping. We may assume that the UV radiation plays a central role in the degradation, and that it probably interacts with the dielectric layers of the PMD above the photodiode area. Then, the electric field role need to be studied, and the PMD dielectrics damage have to be characterized. That is the goal of the next section.

## 3. Study of the Plasma Impact on the Pixel Dielectric Properties

### 3.1. Experimental Set-Up

To better understand the interactions between the plasma and the dielectrics located above the pixel, we deposited, on unpatterned wafers, one or several layers of the studied PMD stack illustrated in Figure 6a, and exposed them to the “strip A” plasma. We used a p-doped substrate at a 1 × 10^15^ cm^−3^ concentration. The samples are described in Table 1. Then, the samples were characterized before and after the plasma exposure by Kelvin probe measurements (Figure 6b) and the COCOS technique (Figure 6c) [3]. Measurement of the silicon photoluminescence intensity in the 900–1300 nm band was also performed [9].

### 3.2. Pre-Metal Dielectrics (PMD) Properties Measurement

#### 3.2.1. Surface Potential Voltage Evolution

Figure 7 summarizes the evolution post plasma exposure of the surface potential *V_CPD_* for the different stacks. Sample 1 does not show a significant potential evolution, meaning that too thin an oxide, lower than 50 nm, does not trap any significant charges during the plasma exposure. All the samples with nitride layer exhibit both negative and positive voltages.

We observe on Figure 8 the evolution of the potential *V_CPD_* at the surface of the top PMD thick oxide. The voltage becomes clearly positive after the plasma exposure at the wafer center. The same evolution is observed on the bottom of the PMD stack, composed of a thin oxide and nitride layers. However, as illustrated in Figure 9, the samples with nitride layers exhibit both polarity voltages whereas the samples with only the oxide layer are showing positive ones.

The potential mapping indicates clearly a non-uniform process. The spot with the more positive potential at the wafer center seems to have an axial orientation which varies between the two wafers. This is linked to the plasma tool hardware, as shows Figure 2b: the “strip A” oxygen plasma is confined in a chamber in which a dipole ring magnet creates a rotating horizontal magnetic field. The axial orientation of the positive potential may be an image of the last orientation of the magnetic field at the end of the process steps.

#### 3.2.2. Silicon Surface Barrier Potential Evolution

The principle of the measurement of the silicon barrier potential *V_sb_* is illustrated in Figure 10. This barrier voltage drops to zero when the p-type silicon holes accumulate, rises to positive value when the p-type silicon is holes depleted, and saturates up to 0.6 V (mid-gap) when the silicon reaches the inversion at the surface. This parameter is from first importance to understand the silicon interface activity, as we will see below.

The *V_sb_* mapping of Figure 11a illustrates that there are enough positive charges in the PMD dielectrics at the wafer center to deplete next inverse the silicon, and in the edge enough negative charges to have an accumulation. The Figure 11b shows the correlation between the silicon surface barrier and the *V_CPD_* voltage across the PMD stack. Globally, the p-type silicon is inverted when *V_CPD_* is higher than 4 V, and accumulated when V_CPD_ is lower than −2 V, and depleted for intermediate V_CPD_ values. But we can notice that some points show a *V_sb_* about 0.6 V, meaning that the silicon is inverted, with a *V_CPD_* of −2 V: this means that a negative *V_CPD_* value is not systematically the proof of a total negative charge in the stack. Figure 10b illustrates that, if we consider different charges plan into the PMD stack, their influences on the total voltage will vary depending of their position: the closer the interface, the lower the influence on the *V_CPD_*. However, due to Gauss theorem, the field driving the silicon surface state is due to the algebraic sum of all the charges into the stack. Finally, *V_sb_* measurement is more accurate than *V_CPD_* to understand the plasma impact on the silicon interface.

#### 3.2.3. Total Charge Measurement

The total charge *Qtot* trapped in the stack is measured using the COCOS technique: charges are deposited on the wafer surface with a corona discharge, up to reach the flat-band voltage in the stack, i.e., a silicon potential barrier equal to zero.

The wafer mapping of Figure 12a confirms that positive charges up to +6 × 10^12^ cm^−2^ appear on the wafer center, whereas negative charges down to −2 × 10^12^ cm^−2^ are observed on the wafer edge. Figure 12b shows here a very good correlation between the *V_sb_* and the total charges *Qtot*, compared to the relatively poor *V_CPD_* correlation of Figure 11b. Here, the silicon inversion is reached with a *Qtot* greater than 4 × 10^12^ cm^−2^, the depletion occurs between 0 and 4 × 10^12^ cm^−2^, and the interface is holes accumulated for negative *Qtot* lower than −1 × 10^12^ cm^−2^.

#### 3.2.4. Silicon Photo Luminescence Signal

The measurement of photoluminescence in the 900–1300 nm range corresponds to the band to band emission in the silicon. The intensity of this signal decreases when the non-radiative recombination increases. Hence, this measurement allows to estimate the density of recombination centers, and thus the intensity of the dark current generation. In our samples, we can assume that the signal is only modulated by the recombination at the interface between the silicon and the dielectric of the sample.

Figure 13a shows that for sample 4, the photoluminescence intensity is high in the center of the wafer and low for some spot at the edge. This result is well correlated with the fact that in the center we have an SiO_2_/Si interface in inversion, highlighted by the *V_sb_* results of Figure 13b, with a low recombination rate. The measured spots with an interface in accumulation show also a high photoluminescence intensity whereas those in depletion, have a weak signal, in good agreement with a previous study [10].

#### 3.2.5. Interface States Density Measurement

The interface density under the PMD stack can be estimated during the COCOS measurement, as a second order variation of the *V_sb_* versus the corona deposited charge. This is similar to impact of the interface states density on the MOS capacitance during a C(V) characterization.

Figure 14 shows that the interface state density under the PMD stack has strongly increased after the plasma exposition. The *D_IT_* wafer dispersion, not represented here, is low.

## 4. Discussion

### 4.1. Interaction Plasma versus Dielectrics

The dielectric stack characterization confirms here that the “strip A” plasma process induces a buildup of trapped charge in the PMD dielectrics layer. Only the nitride layer presence allows both negative and positive charges to be trapped. It has been previously shown [11] that silicon nitride can trap both charges’ polarity by applying an electrical stress via corona charging.

The non-uniformity of the charges mapping across the wafer shows that the electric field induced by the non-uniform plasma also plays a key role in the charge build-up mechanism. Hence, the UV radiation intensity is quite uniformly distributed above the wafer. This means that it cannot explain alone the non-uniformity of the trapped charges.

As a consequence, we propose the following PMD charges generation mechanism, illustrated in Figure 15: the deep UV light generated by the plasma may photo-generate carriers in the oxide and nitride dielectrics, or even break some Si=O or Si–N bonds, due to the strong energy of the UV photons. The generated electrons and holes are then separated by the electric field imposed by the plasma at the wafer surface. With positive voltages, the generated electrons can leak up to the wafer surface and recombine with the plasma charges, whereas the holes stay trapped at the bottom of the nitride layer. This creates a positive charge build-up. With negative voltage imposed by the plasma at the wafer surface, electrons are now blocked in the nitride layers, leading to a final negative trapped charge. This can be similar to what is obtained in silicon-nitride structures exposed to ionizing radiation with polarization applied [12]. Furthermore, band energy configuration as illustrated in Figure 10, helps the charge trapping in the nitride of the PMD dielectric stack. The silicon dioxide is actually a barrier for both electrons and holes.

### 4.2. Relation between the PMD Dielectrics Properties and the Pixel Dark Current

The positive charge in the nitride causes the dark current increase. Indeed, Figure 16 shows a clear correlation between the spatial distribution of the total charges and the value of dark current. A 6 × 10^12^ cm^−2^ charge allows to deplete or even weakly inverse the photodiode pinning p-layer close to the silicon-oxide top interface.

The depletion region extends under the spacer of the gate, or at STI and DTI interfaces. According to the SRH theory [13], dark current generation is maximum at the depleted interface, explaining the large measured dark current. Moreover, the high interface states density observed after the plasma exposure increases again the dark current generated.

The high photoluminescence intensity at the wafer center shows that for samples with a lower p doping, the interface under the positive charge is fully inverted, which blocks the SRH recombination during the measurement. However, the dark current is maximum at the wafer center: this is not consistent with the *Vsb* and photoluminescence results. We can formulate two hypotheses:
First, the p+ pinning implant in the pinned photodiode, which was not present on the COCOS characterization wafers, usually accumulates on the surface and moves the depletion edge deeper into the silicon. But the strong positive charges above the Si surface will move it to depletion, or move the depletion edge closer to the surface. This means that the same positive electric field causing an inversion at the silicon interface on the COCOS wafer may cause a silicon depletion touching the interface on the wafer with the CMOS image sensors. Even if the silicon depletion does not reach the interface, it can be close enough to the surface so that the diffusion length of carriers is longer than the distance from surface to depletion edge as reported in [14]. This strongly enhances the minority carrier diffusion and increases the electron generation from the interface.Finally, even if the interface above the photodiode is inverted, there is always a lateral frontier region close to the photodiode periphery where the inversion will change to depletion, before reaching the accumulation: either close to the STI interface, or under the spacer of the transfer gate. This is illustrated on the pixel TCAD cross section of the Figure 17b,c. Moreover, high electric fields are created by the negative voltage applied on the transfer gate: this can also enhance the dark current generated in the spacer interface area.


### 4.3. Ways to Prevent Plasma Damage on CMOS Image Sensor

We have seen in this study many ways to prevent plasma damage on the pixel photodiode. First, a careful choice of the plasma tools has to be done especially during the final process steps where no forming gas anneal can be used, due to the presence of the organic color resist filters and microlens. It is well known [15] that the forming gas anneal can cure the damage, by de-trapping the dielectrics charges, and passivate the degraded interface states with hydrogen.

During the last process steps, the plasma tools generating both a strong UV radiation and electric field non-uniformity have to be avoided. High density plasma tools with magnetic confinement are in this category. When possible, a process using deported plasma, like strip B, has to be preferred.

Next, some integration scheme can help to reduce the degradation: we have seen the benefits brought by some SiCN layers under the cavity. Another process solution has been proposed in [16] such as increasing the doping of the p+ pinning layer above the photodiode, or increasing the PMD nitride layer conductivity with Si-rich, allowing here a better evacuation or recombination of the UV-generated carriers.

Finally, playing on the photodiode type is also a way to harden the sensor against such damage. For example, it is shown in [16] that p-type pixel collecting holes rather than electrons are much less impacted by such positive PMD nitride charging: this is because the silicon pinning layer is now N-doped, and the positive nitride charges will now re-inforce the electron accumulation. But this p-type pixel will not be immune from negative nitride charging.

## 5. Conclusions

We have seen that an oxygen plasma strip can induce a large increase of the pixel dark current of a FSI CMOS image sensor. We have isolated the deep UV radiation and the non-uniform electric field from the plasma as the main causes of the damage. By characterizing the dielectrics stack after plasma exposure, we identified that positive and negative charges have been trapped during the plasma exposure, mainly in the nitride layer. These trapped charges are modifying the oxide/silicon interface generation activity, as highlighted by the silicon barrier potential and photoluminescence measurement. In particular, the depletion of the top pixel interface induced the large dark current increase. The good correlation between the results obtained on the dielectrics characterization, the measured dark current, and the TCAD simulation mean that the proposed mechanism should be the most probable. Finally, different ways are proposed to prevent such a degradation, by plasma process optimization or with new photodiode integration schemes.

The new industry standard for an advanced CMOS image sensor is now back side illuminated pixels. The different pixel configurations and the new back side interface will give new challenges to optimize the process integration in the goal to reach state of the art low dark current.

## Figures and Tables

**Figure 1 sensors-19-05534-f001:**
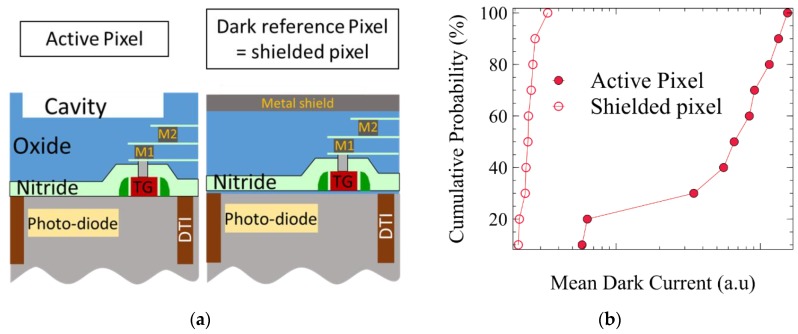
(**a**) Illustration of the configurations of active and dark reference pixel. (**b**) Mean dark current for active pixel of the matrix and dark reference pixel processed with “strip A” process.

**Figure 2 sensors-19-05534-f002:**
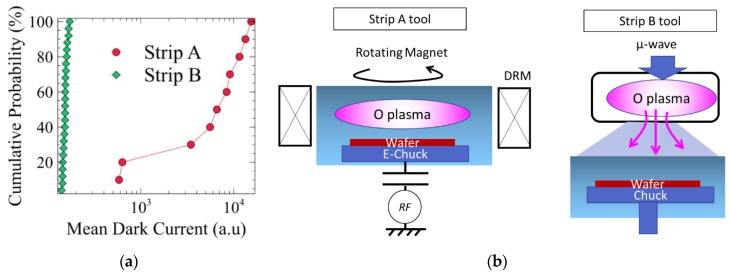
(**a**) Mean dark current for active pixel of the matrix, processed with the two cavity strip processes used in this study. (**b**) “Strip A” is a high-density oxygen plasma created in the cavity etch reactor and “Strip B” is created in a dedicated stripping tool with a remote plasma.

**Figure 3 sensors-19-05534-f003:**
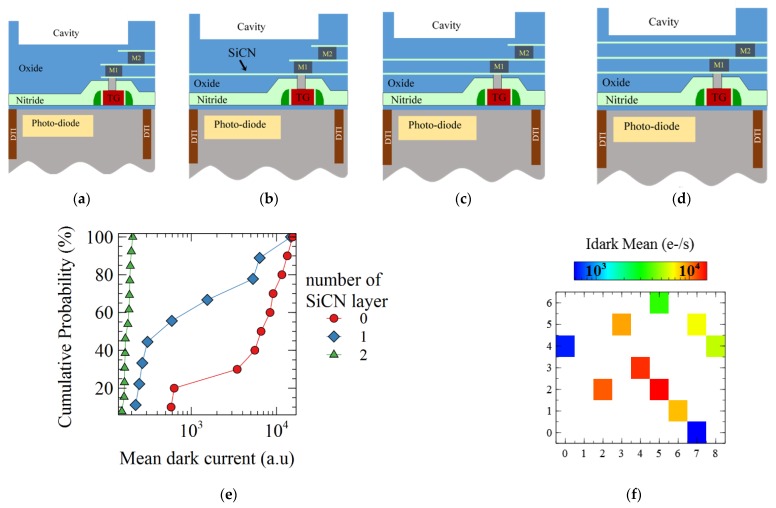
Schematic representation of the pixel cross section is shown for (**a**) the standard process, with one (**b**) two (**c**) or three (**d**) layers of SiCN left above the pixel and under the cavity. (**e**) Mean dark current for active pixel of the matrix with varying number of SiCN layers between the photodiode and the cavity and processed with “strip A”. (**f**) Partial wafer mapping of the mean dark current of pixel with zero SiCN layer and strip A process.

**Figure 4 sensors-19-05534-f004:**
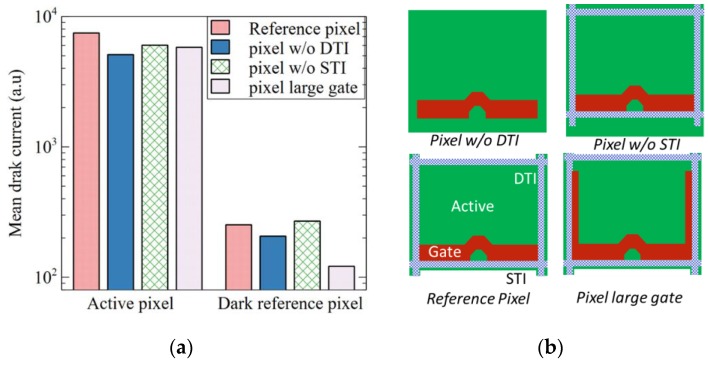
Mean dark current for active pixel of the matrix and dark reference pixel processed with “strip A”. (**a**) Comparison of four pixel versions, the reference pixel with deep trench isolation (DTI) as pixel isolation, the other with implant as pixel isolation, one without STI and one with a larger tranfer gate. The degradation of dark current is the same for all the versions. The pixel layouts are illustrated in (**b**).

**Figure 5 sensors-19-05534-f005:**
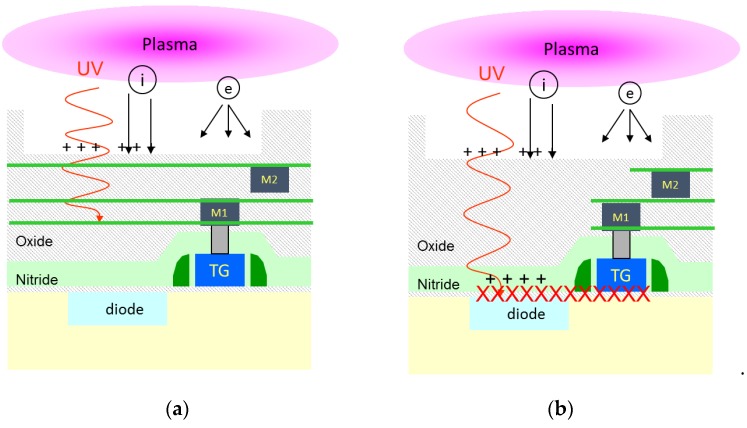
Illustration of the plasma induced degradation mechanism: (**a**) SiCN layers allow to block the ultraviolet (UV) radiation. (**b**) Without any absorption layers on the optical path, the plasma UV associated with the electric field at the wafer surface can lead to charge generation into the dielectrics, and interface states degradation.

**Figure 6 sensors-19-05534-f006:**
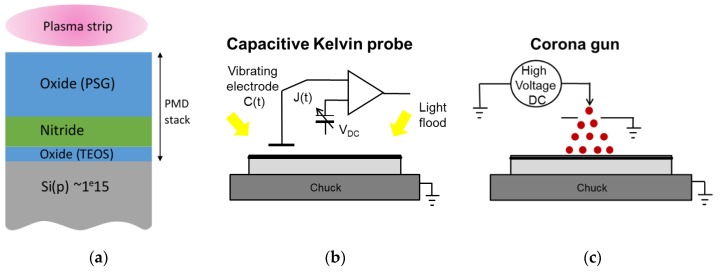
Experimental procedure to study the impact of plasma “strip A” on the dielectric stack. (**a**) Oxide-nitride-oxide of pre-metal dielectric layers. (**b**) Kelvin probe measurement setup. (**c**) Corona method to deposit charge at the wafer surface.

**Figure 7 sensors-19-05534-f007:**
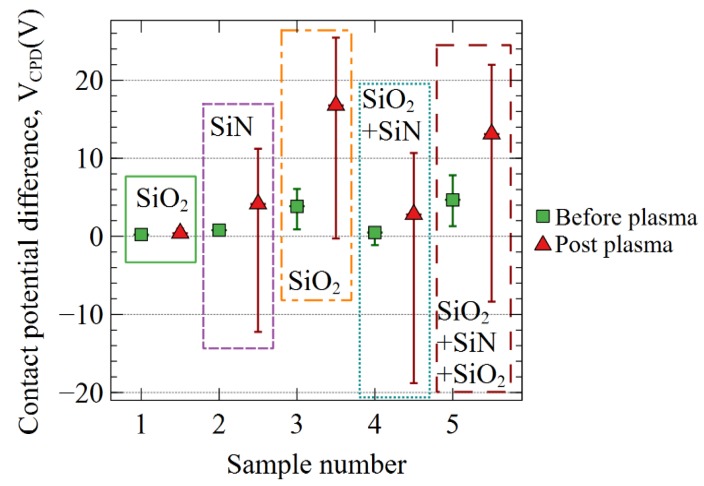
Potential (*V_CPD_*) of the different dielectric stacks as deposited and after plasma exposure.

**Figure 8 sensors-19-05534-f008:**
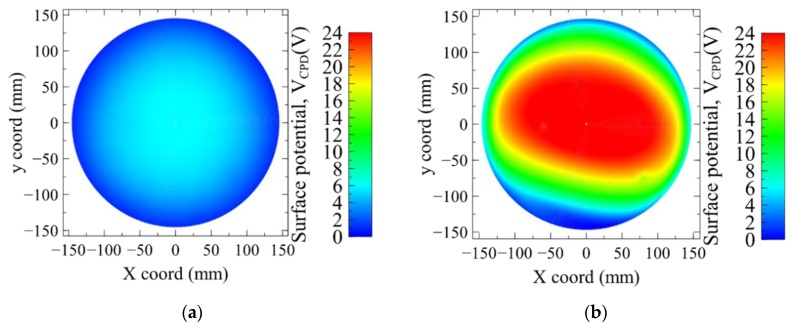
Wafer mapping of *V_CPD_* value for sample 3 with the 500 nm oxide layer. (**a**) before plasma (**b**) after plasma exposure.

**Figure 9 sensors-19-05534-f009:**
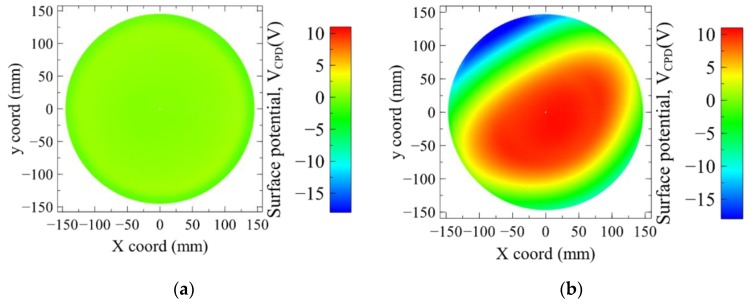
Wafer mapping of *V_CPD_* value for sample 4 with the thin oxide + nitride layers. (**a**) Before plasma (**b**) after plasma exposure.

**Figure 10 sensors-19-05534-f010:**
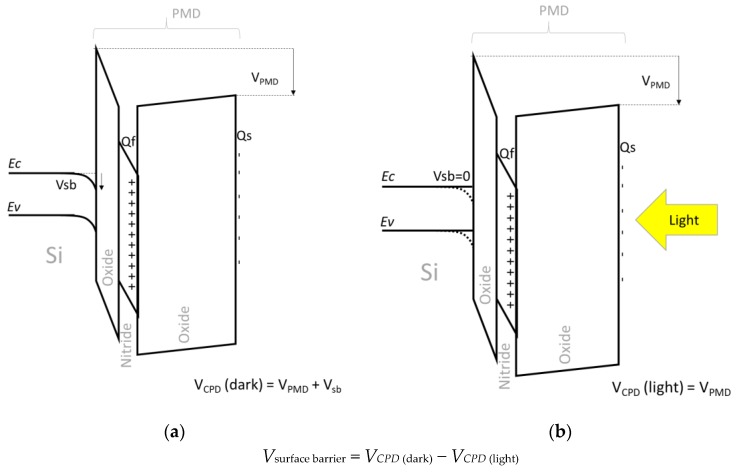
Principle of the silicon surface barrier measurement: (**a**) band diagram in the dark of the oxide + nitride + oxide stack, illustrated with positive charges in the nitride layer *Qf* and few negative charge *Qs* at the top oxide surface. As *Qf* >> *Qs*, the silicon is depleted and a positive *V_sb_* appears; (**b**) the same band diagram under the light, *V_sb_* is now zero due to the photogeneration of electrons and holes in the silicon.

**Figure 11 sensors-19-05534-f011:**
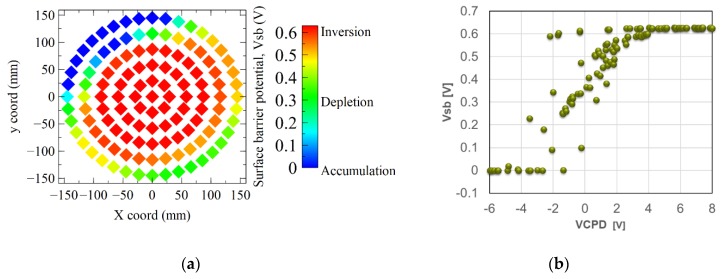
(**a**) Mapping of the surface barrier potential, *V_sb_*, measured on sample 5 with oxide + nitride + oxide layers. (**b**) Correlation between the surface barrier potential and the *V_CPD_*.

**Figure 12 sensors-19-05534-f012:**
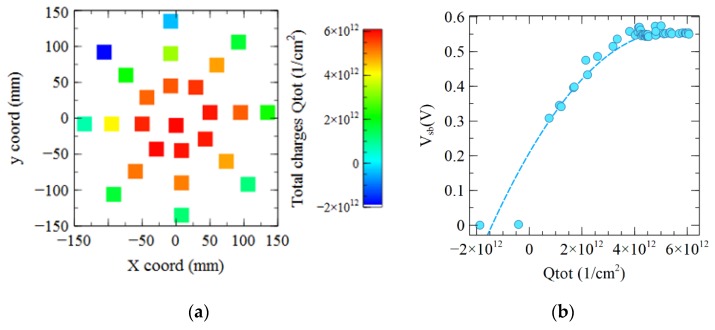
(**a**) Mapping of the total charges measured on sample 4 with oxide + nitride layers. Charges are positive in the center of the wafer and negative at the edge. (**b**) Correlation between the total charge *Qtot* and the *V_CPD_*.

**Figure 13 sensors-19-05534-f013:**
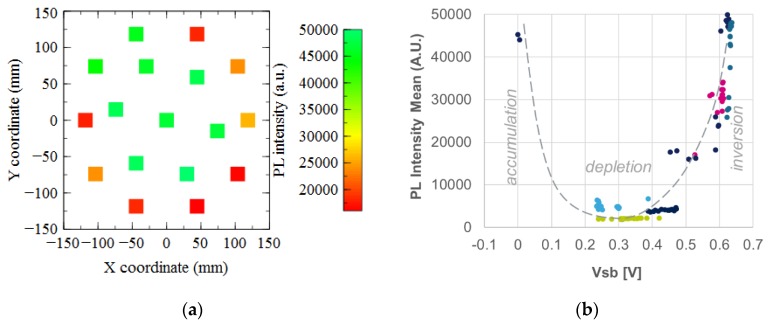
(**a**) Mapping of the photoluminescence intensity measured on sample 4. (**b**) Correlation between the photoluminescence intensity and the silicon surface barrier potential.

**Figure 14 sensors-19-05534-f014:**
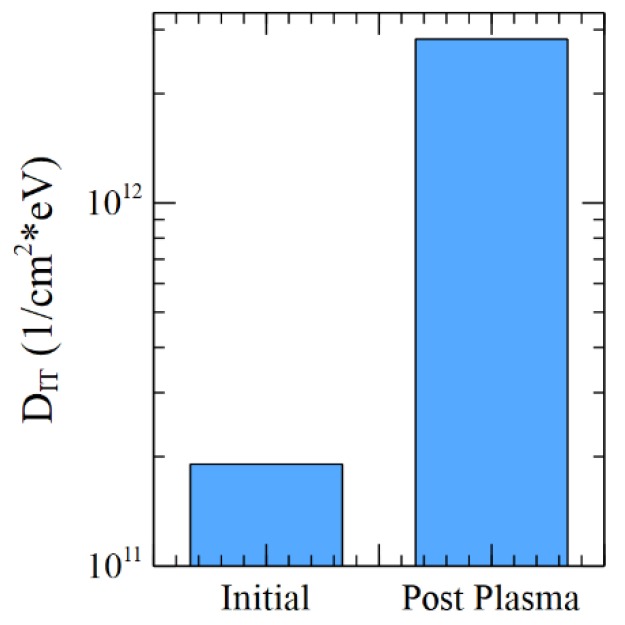
Minimum of the interface states density D_IT_, extrapolated during Qtot measurement on sample 4. Mean value on the wafer.

**Figure 15 sensors-19-05534-f015:**
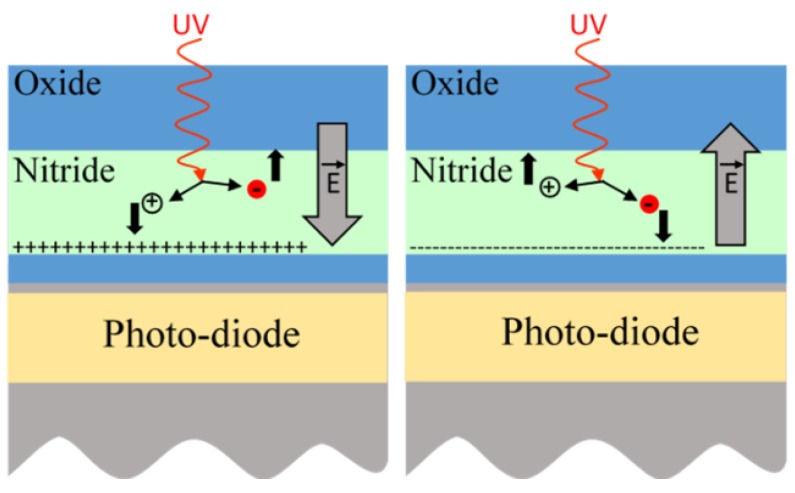
Representation of positive and negative charges creation in the nitride layer of the pixel under plasma exposure.

**Figure 16 sensors-19-05534-f016:**
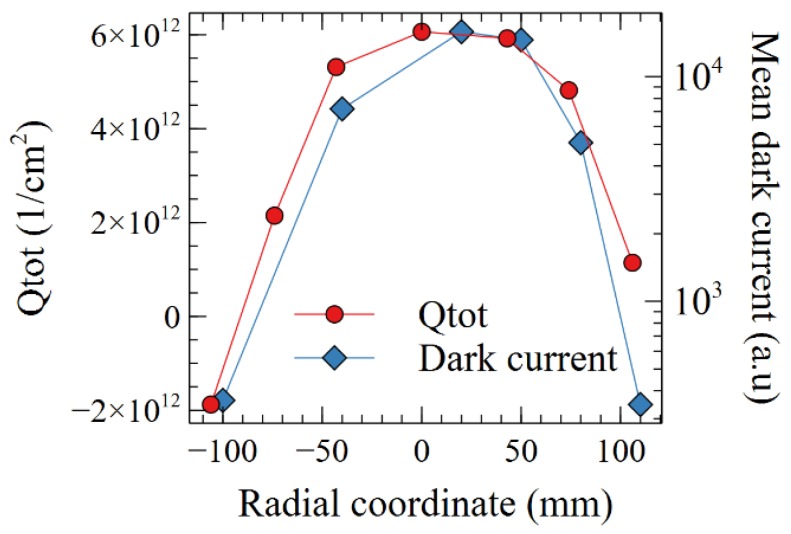
Total charge and mean dark current as a function of the radial coordinate (0 = center of the wafer).

**Figure 17 sensors-19-05534-f017:**
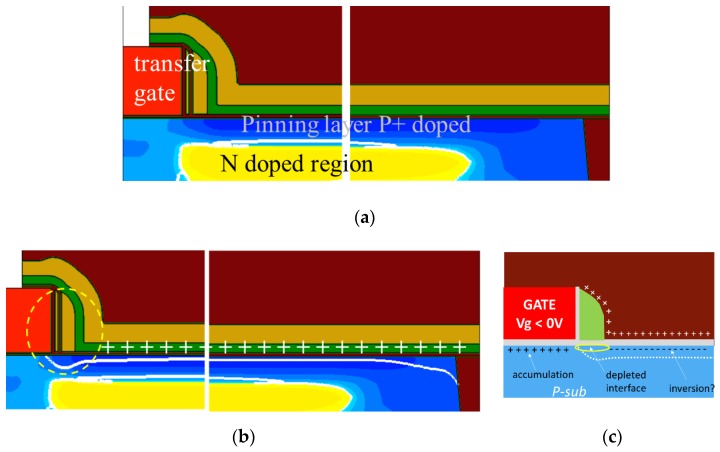
TCAD simulation of the photodiode without (**a**) and with (**b**) fixed positive charge in the nitride. The white line shows the depletion region. (**c**) Illustration of the depleted interface under the spacer, which is the transition area between the accumulation under the transfer gate, and a possible inverted interface above the photodiode.

**Table 1 sensors-19-05534-t001:** Samples description, with the dielectric type and the thickness deposited.

Sample Number	Dielectric Stack
1	Thin Bottom Oxide: about 10–50 nm
2	Anti-Reflective Nitride 50 nm
3	Top Oxide 500 nm
4	Bottom oxide + Nitride 50 nm
5	Oxide + Nitride + Oxide

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
