# Peer review of "CMOS Image Sensors and Plasma Processes: How PMD Nitride Charging Acts on the Dark Current"

_sensors, 2019, doi:10.3390/s19245534_

Round 1

Reviewer 1 Report

Typo's:

L36: Kevin probe should Kelvin probe

L46: SiCN is probably Silicon-Carbon-Nitride instead of Silicon Nitride

In section 2 the order of the figures is confusing. 

General comment: If using the strip B tool completely resolves the issue. Could you explain the relevance of getting full understanding of the issues with strip A tool? I'm assuming that in practice tool B is used for the processing of image sensors.

Overall, your paper looks like a thorough analysis of the problems observed with tool A. I'm assuming the learning may be more widely applicable. 

Author Response

Thank you for your feedbacks:

L36 and 46 typo have been corrected. It is true than setion II contain a lot of figures, we tried to make the legend slightly more clear... You are right, strip B is now the reference process! This non optimized strip A was a good study case, and we actually think that the understanding can be extended to many plasma process using high density plasma source (ie with strong UV lighting), and showing a strong electric non-uniformity.

Reviewer 2 Report

Congratulations with your paper and its overall quality.

Some minor comments:

lines 260-270: the proposed mechanism of charge build-up could have been given further evidence, e.g. if further supported by tests to bring more quantitive data on dependencies of the different parameters in the process, or by refuting other possible mechanisms lines 294-297: this explanation is not completely clear to me. This frontier is the white line in the figure I assume. In case of inversion at the top, there is no additional depletion area that can explain an increased dark current generation. In my opinion, this could however be due to increased electric fields near the surface and transfer gate. 

Author Response

Thank you for your nice feedback:

lines 260-270: It is true that the proposed mechanism of charge build-up stays not really demonstrated here... We had plan to make some partitioning experiments, but unfortunately no time to finish this plan! We previously published in ref [2] one evidence: no dark current degradation was observed when the wafers were exposed to only a deep UV light during the process. That is why we believe that the degradation only occurs when there is both deep UV lighting and electric field. lines 294-297: we add the Fig. 16(c) to illustrate where should occur a depleted interface: it should be on the lateral transition area between the possible inverted interface above the photodiode, and the hole accumulated interface under the gate, as an example. Nevertheless, we agree that the strong electric field brought by the gate negative polarization can also strongly enhance the dark current: we add this explanation on lines 305-307.